

# Behavioural patterns of free roaming wild boar in a spatiotemporal context

Dana Erdtmann[*] and Oliver Keuling[*]

Institute for Terrestrial and Aquatic Wildlife Research, University of Veterinary Medicine Hannover, Hannover, Germany

[*] These authors contributed equally to this work.

## ABSTRACT

Although the almost worldwide distributed wild boar *Sus scrofa* is a well-studied species, little is known about the behaviour of autochthonous, free living wild boar in a spatiotemporal context which can help to better understand wild boar in conflict terms with humans and to find solutions. The use of camera traps is a favourable and non-invasive method to study them. To observe natural behaviour, 60 camera traps were placed for three months in a state forest of 17.8 km$^2$ in the region of the Luneburg Heath in northern Germany. In this area wild boar, roe deer, red deer, wolves and humans are common. The cameras recorded 20 s length video clips when animals passed the detection zone and could be triggered again immediately afterwards. In total 38 distinct behavioural elements were observed, which were assigned to one of seven behavioural categories. The occurrence of the behavioural categories per day was evaluated to compare their frequencies and see which are more essential than others. Generalised Additive Models were used to analyse the occurrence of each behaviour in relation to habitat and activity time. The results show that essential behavioural categories like foraging behaviour, locomotion and vigilance behaviour occurred more frequently than behaviour that "just" served for the well-being of wild boar. These three behavioural categories could be observed together mostly in the night in broad-leaved forests with a herb layer of 50–100%, comfort behaviour occurred mostly at the ponds in coniferous forest. It is also observable that the behavioural categories foraging and comfort behaviour alternated several times during the night which offers the hypothesis that foraging is mostly followed by comfort behaviour. These findings pave the way towards implementing effective control strategies in the wild and animal welfare in captivity.

# INTRODUCTION

Animals behave in order to survive and reproduce themselves (*Naguib, 2006*; *Kappeler, 2009*) and choose different habitats to increase their survival and fitness. Behaviour is defined as control and exercise of movements or signals with which an animal interacts with conspecifics or other components of its animate and inanimate environment, as well as activities which serves for the homoeostasis of an individual (*Kappeler, 2009*). Some animals within a given population, however, will perform much better in some habitats

Corresponding author
Oliver Keuling,
oliver.keuling@tiho-hannover.de

than in others (*Gaillard et al., 2010*). Within a day terrestrial herbivores relocate between foraging areas, drinking and resting sites and places used for other activities at different times of the day (*Owen-Smith, Fryxell & Merrill, 2010*). Predation pressure, inter- and intraspecific competition, diseases and human disturbances can affect the behaviour and consequently the survival and fitness of animals (*Gaillard et al., 2010*). A first step to assess functions of a specific behaviour, and henceforth to analyse behaviour changes, is to watch the behavioural elements performed in specific places at defined times of the day to understand their benefits for survival.

Among the terrestrial even-toed ungulates (*Artiodactyla*) the Suina (*Price, Bininda-Emonds & Gittleman, 2005*; *Gatesy, 2009*) is the only omnivorous non-ruminant suborder with several of original features (*Briedermann, 2009*). Among the Suina the species *Sus scrofa* is distributed almost worldwide (*Lowe et al., 2000*; *Briedermann, 2009*; *Mayer, 2009*). Wild boar are amongst the most intelligent and adaptable large terrestrial mammals in Central Europe (*Briedermann, 2009*) making it very interesting for behavioural analyses in relation to the time of day and different habitat types. Only few studies analysed the behaviour of wild boar under natural conditions (*Allwin et al., 2016*; *Probst et al., 2017*). Most studies were conducted at artificial feeding places (*Schneider, 1980*; *Saebel, 2007*; *Focardi et al., 2015*) or in enclosures (e.g., *Gundlach, 1968*; *Beuerle, 1975*; *Altmann, 1989*) which does not necessarily enable to cover all behavioural elements that would normally occur over the course of a day in a wild population. There is a lack of recent field studies under natural conditions due to the fact that wild boar are widely seen as a pest because of their constant conflict terms with humans, such as crop damage, disease transmission (*Keuling et al., 2013*; *Allwin et al., 2016*; *Probst et al., 2017*) and zoonosis, road traffic accidents, and disturbances to sensitive plant communities (*Maselli et al., 2014*). Though, it is very important to understand the behaviour of wild boar to be able to implement effective management strategies for reduction plans (*Maselli et al., 2014*) as well as for animal welfare in enclosures (*Kovács, Újváry & Szemethy, 2017*).

As the behaviour of wild boar hardly differs from that of domestic pigs (*Stolba & Wood-Gush, 1989*; *GÖT & BAT, 2003*; *Mayer, 2009*), their behaviour can be summarised by: resting, locomotion, behaviour caused by metabolism (ingestion and excretion), comfort, vigilance, social and sexual (*Gundlach, 1968*; *Beuerle, 1975*; *Saebel, 2007*) (see Table 1). Most of the day (70–90%) is spend on foraging to fulfil the animal's basic needs (*Briedermann, 1971*; *GÖT & BAT, 2003*; *Keuling & Stier, 2009*), of which about half is filled by ingestion or locomotion (*Stolba & Wood-Gush, 1989*; *Morelle et al., 2014*). Comfort behaviour, in contrast, is practised much less but serves the important function of well-being (*Keuling & Stier, 2009*).

Wild boar, however, due to their intelligence and adaptability, can learn new attitudes due to training and imitation (*Schneider, 1980*; *Broom, Sena & Moynihan, 2009*; *Sommer, Lowe & Dietrich, 2016*). Studies show that the behaviour of wild boar differs depending on the region (habitat), population, and individual (*Schneider, 1980*). For activity and habitat choice behaviour in particular, the same biotic and abiotic factors are important (*Choquenot, McIlroy & Korn, 1996*; *Lemel, Truvé & Söderberg, 2003*; *Briedermann, 2009*). In general, wild boar prefer broad-leaved forest with older mast species (beech, oak) while

**Table 1  Ethogram for the classification and definition of the behavioural elements of the observed wild boar.**

| Context | Definition |
|---|---|
| **Locomotion (L)** | |
| Walking | Slow movement (pace), every leg is moved at least one step (also backwards possible). |
| Running | Fast movement (trot and faster). |
| Jumping | Jump over an obstacle or ditch. |
| Flight | Abrupt escape from recent whereabouts (optionally just a few steps). |
| **Olfactory behaviour (OB)** | |
| Sniffing | Sniffing on the ground or between plants of the ground and herb layer. |
| Winding | Sniffing in the air or at something (e.g., rubbing tree, camera). |
| Defecating | Emptying of the gut. |
| Urinating | Total drain of the bladder. |
| **Vigilance behaviour (VB)** | |
| Getting frightened | Short wince of the whole body. |
| Pausing | Freeze of motion with alert view and potential additional head lift and look about. |
| Laying down | Young boar presses its body abrupt even on the ground. |
| Guarding | Alert milling around, with lifted head and tail, obvious tense posture. |
| **Foraging behaviour (FB)** | |
| Pawing | Pawing in the ground (e.g., soil, leaves) with a foreleg. |
| Rooting | Rooting in the ground (e.g., soil, leaves) with the snout, also with brushing big branches aside. |
| Salt ingestion | Ingestion of salt at a salt lick by licking, nibbling. |
| Sucking attempt | Young boar attempt to suck on the sow's teats or briefly suck at the standing sow. |
| Suckling | Young boar are suckled by the lying sow. |
| Chewing | Uniform opening and closing of the mouth after foraging (feeding not visible). |
| Feeding attempt | Young boar takes soil/stone into its mouth. |
| Feeding | Ingestion of food with the mouth and chewing afterwards. |
| Drinking | Ingestion of water with the mouth. |
| **Comfort behaviour (CB) –personal hygiene behaviour** | |
| Stretching | Increasing the distance of the hind legs to the forelegs and slightly spreading of the hind legs while simultaneously scuttling with the forelegs. |
| Shaking | Moving its body strongly, briefly and fast back and forth while standing. |
| Rubbing | Rubbing one's body against a tree or another wild boar. |
| Nibbling | Nibbling/rubbing of the open mouth against the rubbing tree. |

**Table 1** (*continued*)

| Context | Definition |
|---|---|
| Scratching | Scratching one's body with the hind legs. |
| Scratching one's bottom | Rubbing one's bottom against the ground while sitting. |
| Rolling | Rubbing one's body against the ground. |
| Wallowing | Laying down (and optionally wallowing) in muddy water. |
| **Social interaction (SI)** | |
| *Active socio negative interaction* | |
| Threating | Keeping another wild boar at distance by threating behaviour. |
| Pushing away softly | Pushing another wild boar softly away with the head, the side of the body or the bottom. |
| Chasing away | A wild boar runs after another wild boar, which departs itself afterwards. |
| Snout knock | A wild boar knocks its head bottom-up in the direction of another wild boar (with/without touching). |
| *Passive socio negative interaction* | |
| Retreating | A wild boar increases the distance to another wild boar, which emitted socio negative behaviour before. |
| *Socio positive interaction* | |
| Nose-to-nose contact | Sniffing at or touching the snout region (being sent of one or both wild boar, also at distance). |
| Nose-to-body contact | A wild boar sniffs at or touches another wild boar with the snout at its body or legs. |
| Playing | Playful behaviour against other wild boar (e.g., exercise fights, apparent copulation attempt). |
| **Sexual behaviour (SB)** | |
| Copulation attempt | A wild boar climbs the bottom of another wild boar. |

foraging (*Berger, 2006*; *Bertolotto, 2010*) which they mainly explore in the first half of their activity time (*Keuling & Stier, 2009*). In contrast, coniferous forest is preferred for their resting sites (*Bertolotto, 2010*) as well as secure places for wallowing and sleeping (*GÖT & BAT, 2003*; *Keuling & Stier, 2009*; *Allwin et al., 2016*). Comfort behaviour often takes place in the second half of the night (*Keuling & Stier, 2009*).

To pave the way of authentic wild boar behaviour in a spatiotemporal context, we aimed to (1) reveal as many behavioural elements of wild boar as possible, and (2) relate functions to them depending to the spatiotemporal occurrence. The following hypotheses are tested: 1. Essential behavioural categories like foraging, vigilance and related locomotion occur more frequently than behaviour serving for the well-being of wild boar like comfort behaviour. 2. Foraging and other related behavioural categories occur in the first half of the night in broad-leaved forest, whereas comfort and related behavioural categories can be observed later in the night in coniferous forest or at places where the animals feel secure.

## MATERIAL AND METHODS

### Study area

The study area "Süsing" was a 17.8 km² state forest located in the Luneburg Heath in Germany. The region is characterised by large-area coniferous forest (*Pinus sylvestris*, *Picea abies*, *Larix decidua* & *L. kaempferi*, *Pseudotsuga menziesii*) and small-area oak (*Quercus robur*) and beech (*Fagus sylvatica*) forests (*Keuling et al., 2013*). The mean annual temperature is 8 °C and the average annual rainfall is approximately 700 mm (*Keuling et al., 2013*). Besides a high number of wild boar (about 8 animals/km² during the study period, calculated according to Rowcliffe via Random Encounter Model (REM) *Rowcliffe et al., 2008*), there are also high numbers of roe (*Capreolus capreolus*) and red deer (*Cervus elaphus*), as well as a few wolves (*Canis lupus*). Verbal permission was provided by Helmut Beuke (head of operations) at the Forestry Office of Oerrel to drive on to the closed forest roads and install the camera traps on their forest property.

### Data collection

Direct observations (compared to radio telemetry) are required to record the behaviour of animals and consequently also get information on activity and habitat choices (*Cagnacci et al., 2010*). One cost-efficient method for the observation of free roaming wild boar is the use of camera traps. The advantage of camera traps is that they are non-invasive (*Rovero, Tobler & Sanderson, 2010*; *Rowcliffe et al., 2011*), and as a consequence, ideal to study nocturnal and crepuscular animals which avoid humans (*Rovero, Tobler & Sanderson, 2010*). The technique is applicable to the study of wild boar given they rarely react to camera traps (*Amelin, 2014*). Using ESRI® ArcGis 10.1, 50 random points (*Rowcliffe et al., 2008*; *Rovero et al., 2013*) separated by a minimum distance of 100 m (*Passon et al., 2012*; *Hofmeester, Rowcliffe & Jansen, 2017*) were determined and afterwards explored with GPS (*Rovero, Tobler & Sanderson, 2010*; *Rowcliffe et al., 2011*). Additionally ten places with a high probability of wild boar occurrence (e.g., wallows, fresh rooting places, salt lick) were selected to reveal all behavioural elements necessitated for the ethogram. Cameras were placed at all 60 places and had an effective detection distance of 8–20 m. Animal's behaviours were able to be defined in distances up to 20–30 m in front of the cameras.

The set-up of the Bushnell® TROPHY CAM™ and Bushnell® TROPHY CAM HD™ camera traps took place on 03.03.2014. The 50 cameras used for statistically evaluable behaviour observations were hung as near as possible to the random points, at trees in 90 cm height, orientated parallel to the ground (*Rowcliffe et al., 2011*; *Meek, Ballard & Fleming, 2012*) to capture some open space on the video clips, and if possible, a deer crossing which comes to or goes away from the camera (*Bengsen et al., 2011*; *Rowcliffe et al., 2011*). The additional ten camera traps, which are statistically irrelevant for the behaviour frequency, were hung at different heights (most of the time higher than 90 cm and with an angle <90° to the ground) depending on the area to capture a large field of view and thus a lot of behaviours. To not disturb the natural behaviour of the animals, no bait or lure were used at the random points (*Rowcliffe et al., 2011*; *Meek, Ballard & Fleming, 2012*). Each camera had a passive infrared sensor (PIR) and recorded, day and night, a 20 s video clip without sound when they were triggered. 1 s after the ending of the latest video the camera traps

could be triggered again (*Rovero, Tobler & Sanderson, 2010*; *Rowcliffe et al., 2011*). The video clips were stored on SD cards, which were changed biweekly. Function of cameras and battery levels were checked during change of SD cards. After about three months, on 04.06.2014, the camera traps were retrieved.

The date and time for each clip was recorded (the time is presented in segments as full hours with the following full hour, e.g., 00:00 o'clock = 00:00:00 - 00:59:59 o'clock).

The habitat was described at each of the camera locations. First, every place was assigned to one of the six types: track, forest aisle, pond (incl. wallows), field edge (simultaneously edge of the forest), salt lick or wooded. After that, within a radius of 10 m the tree and shrub layer were described with main species (no trees/shrubs, broad-leaved, mixed or coniferous forest) and cover (0%, 0-50% or 50–100%). The herb layer was also divided as described. Here the main species were no herbs, common bracken (P.a. = *Pteridium aquilinum*), European blueberry (V.m. = *Vaccinium myrtillus*), bracken and blueberry (P.a.&V.m.) or herbs (e.g., *Rubus* sectio *Rubus*, *Urtica dioica*, Poaceae, a few Cyperaceae and Polypodiopsida). In addition, the cover of deadwood (0%, 0–25% or 25–50%) was registered.

In this study the sampling method "behaviour sampling" and the recording method "time sampling, one-zero sampling" (*Altmann, 1974*; *Geissmann, 2002*) were used. That means, during a sampling interval (video length of 20 s) all visible boar were observed as one group and it was noted for every behavioural element if it occurred in the video clip or not. An ethogram was created following literature review (e.g., *Gundlach, 1968*; *Saebel, 2007*; *Briedermann, 2009*) and own observations, at which exclusively the own observations are shown in Table 1.

## Data analysis

Wild boar could be identified on 1,227 of ca. 8,500 video clips as well as at 57 of 60 places. From 1,169 video clips, a behavioural context could be analysed (645 of 673 video clips at the random points, 524 of 554 video clips at the other ten places), but only the video clips at the random points were statistically analysed, because the other ten did not fulfil the statistical requirements (not randomly, hung at different heights).

To compare the occurrence per day of the seven different behavioural categories at the random points, two analyses were done: First, to calculate the percentage of each behavioural category, the number of observations per behavioural element (BE) and random point (RP) at one day was calculated as the function:

$$N_{\text{obs,d}=1}(BE, RP) = \frac{N_{\text{obs}}(BE, RP)}{d_{RP}}$$

where $N_{\text{obs}}(BE,RP)$ is the total number of observations per behavioural element and random point and $d_{RP}$ is the number of trial days per random point. Then the mean number of observations per behavioural element at one day could be calculated by:

$$\bar{x}(BE) = \frac{\sum(N_{obs,d=1}(BE, RP))}{50}$$

where 50 is the number of random points. The percentage for each behavioural category (BC) in% (with $\sum(P(BC)) = 1$) was then calculated by:

$$P(BC) = \frac{\sum_{(BC)}(\bar{x}(BE)) * 100}{\sum(\bar{x}(BE))}$$

where $\sum_{(BC)}(\bar{x}(BE))$ describes the sum of the mean numbers of observations per behavioural element over all behavioural elements which belong to one behavioural category, and $\sum(\bar{x}(BE))$ describes the sum of the mean numbers of observations per behavioural element over all behavioural elements.

Second, to compare the occurrence of the behavioural categories, the number of observations per behavioural category and random point was calculated as the function:

$$N_{obs}(BC, RP) = \sum_{(BC, RP)} (N_{obs, d=1}(BE, RP))$$

where $\sum_{(BC,RP)}(N_{obs,d=1}(BE,RP))$ describes the sum of the numbers of observations per behavioural element and random point at one day over all behavioural elements, which belong to one behavioural category. Afterwards pairwise comparisons of means (over all random habitats, $N = 300$) were conducted with R software version 3.1.1 (*R Core Team, 2014*), using the packages "nlme" (*Pinheiro & Bates, 2014*) and "multcomp" (*Hothorn, Bretz & Westfall, 2014*). Therefore, the linear mixed model (LMM) (*Dormann & Kühn, 2009*) combined with the post hoc analysis least squares means (LSMEAN) (*SAS Institute Inc., 2011*) with Tukey adjustment (*NIST/SEMATECH, 2013*) was performed.

For the analyses of the behaviour in a spatiotemporal context, similar behavioural elements were grouped as listed: locomotion; sniffing and winding; defecating and urinating; vigilance behaviour; rooting and pawing; salt ingestion; sucking attempt and suckling; chewing and feeding (attempt); drinking; wallowing, nibbling and stretching; shaking; rubbing; scratching (one's bottom) and rolling; social interactions; sexual behaviour (see Table 1). For each grouping, the number of video clips per time of day was summed over all 60 camera locations with Microsoft® Excel 2007 to determine the activity maxima in general. Significant habitat preferences per behaviour were derived from a generalised additive model (GAM) dependent on the time of day and habitat type. Using the data from the random camera locations, it was calculated with R software version 3.1.1 (*R Core Team, 2014*), using the "mgcv" package (*Wood, 2014*), for each behaviour with greater than 20 observations. For the same data, tests for spatial dependence of residuals were conducted on a sample of 1,000 observations. We calculated Moran's I for the first lag with R software version 4.0.2 (*R Core Team, 2020*), using the "ncf" package (*Bjornstad & Cai, 2020*). We did not find significant spatial autocorrelation. Since it is not possible to monitor the whole study area completely and consequently every possible habitat type, we can just draw conclusions out of the results given by random placed camera traps.

## RESULTS

Comparing the proportion of the six observed behavioural categories at the 50 random points, locomotion accounts for more than half of the observations (52%). This behaviour

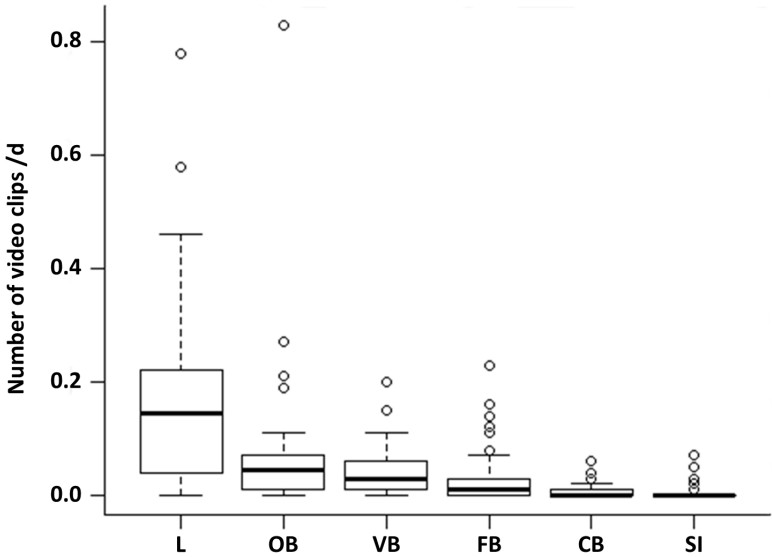

**Figure 1** **Occurrence of the behavioural categories at the random points.** The mean number of videos clips per day is shown for the six behavioural categories (L = locomotion, OB = olfactory behaviour, VB = vigilance behaviour, FB = foraging behaviour, CB = comfort behaviour, SI = social interaction) as box plots with minimum, lower quantile, median, upper quantile, maximum and outlier, observed at the random points (N = 1407; 645 videos clips).

occurred significantly more often than all other behavioural categories (Fig. 1, LMM & LSMEAN see Table 2). It was followed by olfactory (22.02%), vigilance (13.33%) and foraging behaviour (8.81%). The olfactory behaviour occurred significantly more often than foraging behaviour and comfort behaviour as well as social interactions. Vigilance behaviour occurred significantly more often than comfort behaviour and social interactions. Comfort behaviour (1.99%) and social interactions (1.85%) were rarely observed. There were no significant differences between all other pairwise comparisons.

The ten non-random cameras were additionally used for general descriptions of behavioural elements that only occurred there: salt ingestion, feeding attempt, getting frightened, stretching, nibbling, wallowing, chasing away, snout knock, and copulation attempt. The observed wild boar are crepuscular and nocturnal because their main activity was between 17:00 and 08:59 o'clock.

The activity maxima of locomotion occurred in the hour of 22:00 o'clock and in the hour of 03:00 o'clock. During this time the wild boar significantly avoided tracks and significantly preferred forest aisles, ponds and broad-leaved forest with 50–100% herbs and 25–50% deadwood (GAMs see Appendix S1).

The highest activity maximum of sniffing and winding (olfactory behaviour) was between 20:00 and 21:59 o'clock and a secondary maximum in the hour of 03:00 o'clock. During this time the wild boar significantly avoided tracks and habitats with a shrub layer out of coniferous forest (GAMs see Appendix S1). Ponds and habitats with 50–100% herbs and 25–50% deadwood were significantly preferred. Data show no obvious tendency for defecating and urinating.

**Table 2 Results of the LMM and LSMEAN.** Comparison of the occurrence of each behavioural category with each other. The estimate and *p*-value of each pair wise comparison of means with Tukey adjustment is shown.

| Pair wise comparison | Estimate | *p*-value |
|---|---|---|
| Locomotion–Olfactory behaviour | 0.084 | <0.001*** |
| Locomotion–Vigilance behaviour | 0.106 | <0.001*** |
| Locomotion–Foraging behaviour | 0.120 | <0.001*** |
| Locomotion–Comfort behaviour | 0.141 | <0.001*** |
| Locomotion–Social interaction | 0.141 | <0.001*** |
| Olfactory behaviour–Vigilance behaviour | 0.023 | >0.05 |
| Olfactory behaviour–Foraging behaviour | 0.037 | 0.007** |
| Olfactory behaviour–Comfort behaviour | 0.057 | <0.001*** |
| Olfactory behaviour–Social interaction | 0.057 | <0.001*** |
| Vigilance behaviour–Foraging behaviour | 0.014 | >0.05 |
| Vigilance behaviour–Comfort behaviour | 0.034 | 0.014* |
| Vigilance behaviour–Social interaction | 0.035 | 0.012* |
| Foraging behaviour–Comfort behaviour | 0.020 | >0.05 |
| Foraging behaviour–Social interaction | 0.021 | >0.05 |
| Comfort behaviour–Social interaction | <0.001 | >0.05 |

The vigilance behaviour had an activity maximum at 22:00 o'clock and a secondary maximum at 03:00 o'clock. During this time wild boar significantly preferred forest aisles, ponds, a tree layer out of broad-leaved and mixed forest, a shrub layer out of broad-leaved forest, a herb layer with a cover of 50–100% and 25–50% deadwood (GAMs see Appendix S1).

The foraging behaviour had an activity maximum at 17:00 o'clock for salt ingestion, which only occurred at the salt lick, and for chewing and feeding (attempt), which occurred significantly more often at ponds and places with a shrub cover of 50–100% and blueberries (GAMs see Appendix S1). At 19:00 o'clock, there was a secondary maximum again for chewing and feeding (attempt). In the hour of 21:00 o'clock a maximum of salt ingestion and for water intake (only at ponds) was observed. Another maximum occurred at 22:00 o'clock for sucking attempt and suckling, which mostly occurred at the ponds and at the salt lick, and for rooting and pawing. Pawing could significantly be observed at forest aisles, ponds and broad-leaved forest with herbs (GAMs see Appendix S1). At 03:00 o'clock there was another maximum for rooting and pawing and in the hour of 04:00 and 06:00 o'clock again two low secondary maxima for chewing and feeding (attempt).

The comfort behaviour showed a secondary maximum for shaking in the hour of 17:00 o'clock at the salt lick. Furthermore, at around 20:00 o'clock there was a maximum for all elements of comfort behaviour, e.g., for shaking, which mostly occurred at the ponds this time. Another secondary maximum was in the hour of 23:00 o'clock for wallowing, mostly followed by rubbing, nibbling and stretching, which occurred only at the ponds. In the hour of 00:00 o'clock there was a secondary maximum for scratching (one's bottom) and rolling, in which scratching often occurred at the ponds while rubbing.
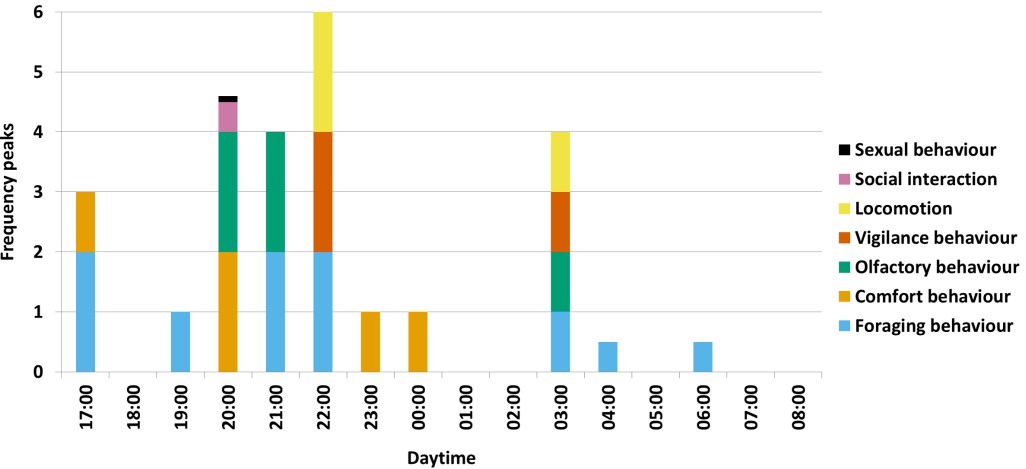

**Figure 2** **Activity maxima per behavioural category.** A visual summary of the results is shown for all seven behavioural categories with maxima (= two frequency peaks), secondary (= one frequency peak) and low maxima (a half frequency peak, respectively a tenth frequency peak for sexual behaviour) (with e.g., 00:00 = 00:00-00:59) (N = 1227).

The social interactions had a low maximum (compared to the size of the maxima of the other behavioural categories) in the hour of 20:00 o'clock. In general, this behavioural category occurred more often in the first half of the night with preferred habitats of forest aisles, ponds, a shrub layer out of 0–50% broad-leaved and mixed forest, herbs and 25–50% deadwood (GAMs see Appendix S1). The sexual behaviour only occurred once at the salt lick in the hour of 20:00 o'clock.

For the several activity maxima per behavioural category in total, the olfactory behaviour occurred mostly in the hour of 20:00 o'clock in form of winding at a rubbing tree during comfort behaviour, in the hour of 21:00 o'clock in form of winding at the salt lick during salt ingestion and in the hour of 03:00 o'clock in form of sniffing on the ground during rooting (Fig. 2).

Furthermore, foraging behaviour and comfort behaviour alternated during the night. After awakening, wild boar first attended to foraging behaviour between 17:00 and 19:59 o'clock, followed by a short maximum of comfort behaviour in the hour of 20:00 o'clock. Afterwards, between 21:00 and 22:59 o'clock, the animals again attended to foraging behaviour until a longer period of comfort behaviour can be observed between 23:00 and 01:59 o'clock. To a minor degree, the rest of the night (02:00-08:59 o'clock) is used for foraging behaviour.

## DISCUSSION

During the observation of wild boar with camera traps, 38 behavioural elements were observed in this study which could be combined into seven behavioural categories. The behavioural category locomotion occurred the most in this study, followed by olfactory, vigilance and foraging behaviour. In many other studies (e.g., *Stolba & Wood-Gush, 1989*; *GÖT & BAT, 2003*; *Saebel, 2007*) foraging behaviour was the most observed behavioural

category. A reason for this might be that in these studies locomotion was always analysed in its pure form and not when it occurred together with other behavioural categories like foraging behaviour (*Briedermann, 1971*). Another reason might be that in our study the duration of the different behavioural categories were not measured and it could be that camera traps are biased towards faster movements (*Rowcliffe et al., 2016*) like running. However, fast locomotion (i.e., running and flight) accounts for only 22.27% of the locomotion in total and 11.58% of all observations. In addition, wild boar never move fast for longer time spans (*Briedermann, 2009*; *Morelle et al., 2014*; *Keuling et al., 2018*), therefore, fast locomotion will not have strong influence on the results. On the other hand, slow behaviours such as foraging take longer and might therefore result in multiple videos captured by the same camera trap. It is also possible that other studies counted sniffing for food as foraging behaviour, which was also often seen in this study. It is important to note, however, that related studies focused on domestic pigs (e.g., *Stolba & Wood-Gush, 1989*; *GÖT & BAT, 2003*; *Mayer, 2009*), wild boar living in enclosures (e.g., *Gundlach, 1968*; *Beuerle, 1975*; *Altmann, 1989*) or observed at feeding places (e.g., *Schneider, 1980*; *Saebel, 2007*; *Focardi et al., 2015*) and hence might show difference to behaviour in the wild. Furthermore, we observed only one wild boar population and our study period amounted just a quarter year and does not reflect the average for an entire year. Wild boar in this study spent more time to foraging then undertaking comfort-related behaviour. According to other studies personal hygiene contributes less to the basic need of wild boar compared to foraging, because the latter serves to ensure survival and personal hygiene "just" for well-being (*Saebel, 2007*; *Keuling & Stier, 2009*). Consequently, our results and that from other studies confirm the hypothesis that behavioural categories, which are essential for survival like locomotion, vigilance and foraging behaviour, occur more often than categories serving for the well-being. Since olfactory behaviour occurred together with essential behaviour and those serving for the well-being it is not clearly assigned to one of them.

## Behaviour in a spatiotemporal context

The maxima of locomotion and vigilance behaviour were observed with foraging behaviour (Fig. 2). Wild boar have to travel long distances while foraging and often have to cross open and unsecure spaces (*Meynhardt, 1982*; *Cahill, Llimona & Gràcia, 2003*), hence vigilance behaviour to avoid predators is important. Meanwhile the observed wild boar mostly used forest aisles or stayed in broad-leaved forests with a herb layer of 50–100%. In other studies it was found that wild boar preferred broad-leaved forest for foraging (*Berger, 2006*; *Bertolotto, 2010*). Wild boar move fast and take the shortest path when crossing an open unsecure space (*Meynhardt, 1982*). Manmade forest aisles that are rarely used by humans are probably used by wild boar (*Allwin et al., 2016*) to allow fast movement through forest areas. Thus, the hypothesis that foraging and related behaviour occur in broad-leaved forest is confirmed.

Social interactions and the only observation of sexual behaviour occurred during the maximum of comfort behaviour. This may be because comfort behaviour (*Saebel, 2007*) and social interactions could be observed mostly at the ponds (containing wallows)

where the animals feel safe (*Keuling & Stier, 2009*). In addition, all ponds were located in coniferous forest, and wild boar prefer pine trees for rubbing (*Mayer, 2009*). Furthermore, these three behavioural categories could be observed many times at the saltlick. In general, however, nearly all of the seven behavioural categories could be observed at the ponds. Consequently, the hypothesis that comfort and related behaviour occur in coniferous forest and at places where the wild boar feel safe cannot be rejected.

If we compare the alternation of foraging and comfort behaviour during the night with the results of another study (*Gundlach, 1968*), the observations of the other study lack the first period of foraging behaviour after awakening and they also refer to diurnal wild boar. Our data support the results of other studies (*Saebel, 2007*; *Keuling & Stier, 2009*) which found that wild boar mostly attend to foraging behaviour in the first half of the night while a higher occurrence of comfort behaviour during the second half of the night is not obvious. It rather gives the impression that foraging is always followed by comfort behaviour. Consequently, comfort behaviour occurs later in the night than foraging behaviour.

## Functions of the behavioural elements

The behaviour of an animal essentially contributes to its survival and reproductive success (*Naguib, 2006*; *Kappeler, 2009*). If we generalise the ecological model for the locomotion of wild boar (*Morelle et al., 2014*), it appears that the behaviour of wild boar is a result of the interaction of intrinsic (energy gain, escape from predators and/or conspecifics, reproductive success) and extrinsic (habitat, climate, presence of predators) factors - and thus, it is the struggle of wild boar with its biotope (*Naguib, 2006*). Our study supports this hypothesis. Further, we distinguished between basic animal behaviour serving the survival of the individuals and the sounder, and comforting behaviour aimed at the well-being of the individuals.

Our data supports, that wild boar use different behavioural elements for reaching different food resources. For example, rooting and pawing serve for the exposing of food sources in the ground (*GÖT & BAT, 2003*). Wild boar can distinguish between food places of different quality and relocate them which saves energy and time (*Held et al., 2005*). Moreover, sows suckle their offspring and therefore invest in the breeding and survival of their offspring (*Vetter et al., 2016*).

Our results show, that the functions of different behavioural elements are closely related. Wild boar, for example, have a very developed sense of smell (*Graves, 1984*; *Mayer, 2009*). The olfactory behaviour serves for foraging and avoidance of predators (sniffing and winding) as well as for intraspecific communication by defecating and urinating. Also rubbing, nibbling as well as nose-to-nose contact and nose-to-body contact serve for intraspecific communication.

Vigilance behaviour (pausing) seemed to be a reaction to the camera traps. Our results show that wild boar, compared to other animals, hardly react to camera traps, but when they react, they do it by eye-contact or pausing (*Amelin, 2014*). Wild boar are reclusive animals (*Gundlach, 1968*; *Beuerle, 1975*; *Altmann, 1989*). Vigilance behaviour is used by wild boar to avoid predation (e.g., by humans or wolves), for example when a sow guards a glade before other sows and young animals follow her. When pausing or laying down, the

movement is abruptly stopped which otherwise would produce a noise, which predators could hear. Moreover, young boar are very camouflaged while laying down due to their striped pattern (*Briedermann, 2009*). The animals also use this moment to scan their environment multisensory (*Quenette & Desportes, 1992*). If the boar do not find the source of the noise or sense disturbing them, it could be that they react with flight.

The behavioural category comfort behaviour mostly serves for two functions, personal hygiene behaviour and resting behaviour. Looking at the personal hygiene behaviour, wild boar use wallowing for thermoregulation because they are not able to sweat and a mud layer also keeps stinging insects away (*Meynhardt, 1982*; *GÖT & BAT, 2003*; *Briedermann, 2009*). According to another study, wild boar immobilise stinging insects with help of the mud and afterwards remove them by rubbing and similar behaviour (*Mayer, 2009*). Rubbing is also caused by hair change in spring-time (*Briedermann, 1971*). Thus, comfort behaviour serves for the well-being of the animals in general (*GÖT & BAT, 2003*). In contrast to the results of previous studies, where stretching was always observed after resting behaviour (*Briedermann, 2009*), in our study stretching also could be seen three times mostly after rubbing and before shaking.

The social interactions of wild boar have different functions. Nose-to-nose contact and nose-to-body contact serve as intraspecific communication (see above). This is important for the mother-infant-relationship (*Gundlach, 1968*; *Meynhardt, 1982*), and for sexual behaviour (e.g., courtship of boars, boar fights), which serves for reproduction. It is said that each behaviour is noticed by group members and has social consequences (*Stolba & Wood-Gush, 1989*), allowing them to learn from each other (*Schneider, 1980*; *Briedermann, 2009*; *Morelle et al., 2014*). Young boar train from an early age on fighting and copulation in a playful manner (*Gundlach, 1968*; *Meynhardt, 1982*; *GÖT & BAT, 2003*), which they use later during the mating season for boar fights and mating. Wild boar also compete for food, however, they have a stable food hierarchy (*Beuerle, 1975*; *GÖT & BAT, 2003*; *Saebel, 2007*) to avoid unnecessary competition and to save energy.

Resting behaviour like sleeping was not observed in this study. Wild boar rest at their daytime resting sites (*Gundlach, 1968*; *Meynhardt, 1982*) which were never placed in front of any of the 60 camera traps. As wild boar prefer dense vegetation for their resting places (*Allwin et al., 2016*) it is statistically unlikely to catch such places randomly, since the camera traps need some open space to work correctly (cf. data collection). Again, we also do not know any resting place of wild boar in our study area, consequently it was not possible to place one of the 10 additional camera traps at their daytime resting sites. To analyse this behaviour in following studies we suggest permanently placing recording video systems at preferred resting sites which should be determined before with help of telemetry (*Lampe, 2004*; *Sándor et al., 2014*). Since our results stem only from videos in forest habitats, a lack of observations from open areas may explain lacks of activity maxima in the hour of 18:00 o'clock and between 01:00 and 02:59 o'clock, because at that time wild boar were probably on greens, fields or at baiting stations (in surroundings of private hunting grounds) for foraging. Another possibility is that the animals had an activity break between 01:00 and 02:59 o'clock in which time resting behaviour could have been observable. It has already

been suggested that free roaming wild boar have a rest period in the second half of the night (*Briedermann, 1971*), diurnal wild boar around midday respectively (*Allwin et al., 2016*).

The expansion of humans results in wild boar's habitat reduction. Due to the lack of natural predators in many places and increasing food supply, the wild boar population numbers are constantly increasing (*Massei et al., 2014*) and consequently it comes to their invasion into urban areas (*Kotulski & König, 2008*; *Toger et al., 2018*; *Conejero et al., 2019*). This leads to many conflicts between wild boar and humans. As wild boar are very adaptable, one method alone is not sufficient to reduce the animal's number. In addition to the procedures already known (cf. *West, Cooper & Armstrong, 2009*; *Tack, 2018*), the results of this work show further possibilities, such as hunting the animals during their activity times at night with night vision devices at known social locations, and avoiding additional foraging resources (e.g., access to food waste) during their foraging activity times. In general, knowledge of habitat preferences and behavioural needs are useful for habitat management. Keeping wild boar in their "comfort habitats" could reduce human wild boar conflicts such as crop and rooting damages, if enough preferred habitats are available. Additionally, the public needs to be better informed about the effects of increasing wild boar population numbers, as there is, for example, a growing negative public opinion towards hunting (*Tack, 2018*). Consequently, wild boar behaviours drive the human perception of the wildlife-human conflict and thus determine the way of implementing wildlife population management measures. But we are also responsible for the living conditions of wild boar in captivity such as zoos and domestic pigs in factory farming. Enclosure design and activities can significantly improve animal welfare. Our results can be used as a model to show which habitat requirements an enclosure should fulfil (e.g., coniferous woods for rubbing, wallows, retreats), at what times and in what form food should be given (rooting possibilities), and when the animals should be allowed to rest. These are only a few examples.

The behavioural elements salt ingestion, feeding attempt, getting frightened, stretching, nibbling, wallowing, chasing away, snout knock, and copulation attempt could only be observed at the non-random points and could therefore not be statistically analysed. On the other hand the behavioural elements defecating, urinating, guarding, suckling, scratching one's bottom and rolling only occurred at the random points. Many important behavioural elements like wallowing and rubbing occur only in certain places and are not necessarily detected by a random distribution. Therefore, it is even more important to observe not only random places but also known whereabouts of the wild boar in order to uncover the entire behavioural repertoire of the species and to describe their needs. In order to increase the chance for documentation of rarely observed behaviours, further studies should be conducted in which more camera traps are placed comparing different localities and populations, as several behaviours could not be observed in this study due to the biotope (habitat) of animals. Furthermore, mating and mating-related fights of males take place from November till January (*Meynhardt, 1982*; *Altmann, 1989*; *Briedermann, 2009*) which is beyond the observation season. The season also has an influence on biotope choice (*Keuling, Stier & Roth, 2009*). Thus in future studies, it would be advisable to observe wild boar for at least one year via camera traps to get a whole impression of their spatiotemporal

behaviour. This year long observation would also account for possible weather influences on the activity and habitat choice of wild boar (*Saebel, 2007*; *Briedermann, 2009*; *Allwin et al., 2016*). Sound recordings could, hence, be taken when looking at courtship interactions to record communication behaviour and to eliminate the influence of the data collection via camera traps.

Currently, the numbers of wolves are rising across Europe (*Randi, 2011*; *Arbieu et al., 2019*) and hence, likely influence the behaviour of wild boar, like they do in other species, e.g., roe deer (*Bongi et al., 2008*) and alpine ibex (*Grignolio et al., 2019*). This study can serve as a baseline study to record behavioural changes of wild boar in areas in which apex predators are recurring and increasing. To see if the spatiotemporal behaviour changes, future studies could compare different study areas (including or excluding predators, hunting and other human impact, different habitats, different seasons during some consecutive total years).

## CONCLUSION

The behaviour of wild boar is a result of the interaction of intrinsic and extrinsic factors—and thus, it is the struggle of wild boar with its biotope. Essential behavioural categories like foraging behaviour, locomotion and vigilance behaviour occurred more frequently than behaviour "just" serving for the well-being of wild boar. Accordingly, the activity maxima of these three behavioural categories could be observed at the same time and predominantly in the first half of the night. To suggest some management measures, during this time the hunting pressure should be enlarged and the supply of human food resources should be avoided. Additionally, the results of this study are an important contribution towards wild boar welfare in enclosures, showing their basic requirements for habitats to fulfil their natural behavioural repertoire. Video traps are a good method to observe the behaviour of animals under natural conditions. Although video traps are not always reliably triggered by wild boar, using a high number of them gives an effective alternative compared to telemetry which would require wild boar disturbing direct observations. In further studies it would be advisable to observe wild boar year round with additional sound recordings to get an overall impression of the wild boar behavioural repertoire and to increase the chance of detecting rare behaviours as well as behavioural changes due to human or recurring large predator impacts.

## ACKNOWLEDGEMENTS

We would like to thank all people, who helped within our research: all hunters and employees of the Forestry Office of Oerrel and the Big Game Association "Hochwildring Süsing", all colleagues and students for help during field work and analyses, and Katrin Ronnenberg for statistical support. We are grateful to Taren Heintz and Marie Sange for revising the English and to Maraja Riechers for reviewing an early draft of the manuscript.

### Funding

This study was supported by means of the ''Jagdabgabe'' (hunting taxes) of the Federal Ministry of Nutrition, Agriculture and Consumer Protection of Lower Saxony/Germany. There was no additional external funding received for this study. The funders had no role in study design, data collection and analysis, decision to publish, or preparation of the manuscript.

### Grant Disclosures

The following grant information was disclosed by the authors:
Federal Ministry of Nutrition, Agriculture and Consumer Protection of Lower Saxony/Germany.

### Competing Interests

The authors declare there are no competing interests.

### Author Contributions

- Dana Erdtmann performed the experiments, analyzed the data, prepared figures and/or tables, authored or reviewed drafts of the paper, and approved the final draft.
- Oliver Keuling conceived and designed the experiments, analyzed the data, authored or reviewed drafts of the paper, and approved the final draft.

### Animal Ethics

The following information was supplied relating to ethical approvals (i.e., approving body and any reference numbers):

Due to the German law for animal welfare no approval was needed for involving vertebrate animals in a non-invasive observation method by video-traps in free roaming vertebrates.

### Field Study Permissions

The following information was supplied relating to field study approvals (i.e., approving body and any reference numbers):

Verbal permission was provided by Helmut Beuke (head of operations) at the Forestry Office of Oerrel to drive on to the closed forest roads and install the camera traps on their forest property.

### Data Availability

The raw measurements are available in the Supplementary File.

### Supplemental Information

Supplemental information for this article can be found online at http://dx.doi.org/10.7717/peerj.10409#supplemental-information.

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
