# Peer review of "Behavioural patterns of free roaming wild boar in a spatiotemporal context"

_PeerJ, doi:10.7717/peerj.10409_

## Round 0.1 · original submission · Major Revisions

The reviewers provided extensive and detailed comments, including request of data re-analyses. Both reviewers specify the lack of discussion on the implications of the results on management strategies for the wild bore populations.

Reviewer 1 ·

Basic reporting

no comment

Experimental design

Two major concerns about the experimental design: i) only one study area and sampling season, and ii) the frequency estimated by each behavior could be biased due to movement related questions (see comments to authors for further details).

Validity of the findings

no comment

Additional comments

The manuscript is focused in the description of the wild boar behavior s, and their frequency and spatio-temporal patterns, in a natural population. The interest in wild boar is nowadays increasing in Europe mostly due to its role in growing conflicts with human activities, such as those linked to shared diseases (e.g., African swine fever) and agricultural damages, among others. However, even when the topic is relevant and timely, I have some major concerns that should be addressed in a new version of the manuscript.

i) One is in relation to the sampling design just focused on a single population and only one sampling season. In my opinion this is not the best design to describe the ethogram of a species since at least some additional localities would be needed and, ideally, a full year should have been monitored. Some of these drawbacks were discussed in the previous version, but all of them should be considered to tone-down some sentences describing the aims (Line 86) and gain precision when interpreting results in the new version of the manuscript.

ii) A major concern is related with the comparison among behaviors' frequency. Some authors have pointed that detectability of behavior from camera traps can be biased and slow movements can be underrepresented (e.g. foraging) and overrepresented others (e.g. locomotion) (e.g. Rowcliffe et al., 2016 – Remote Sensing in Ecology and Conservation 2: 84-94; Caravaggi et al., 2020 – Conservation Science and Practice e239). Likely, this fact is affecting the results reported here, since behaviors involving faster movements were those with higher rates in video captures (e.g., locomotion > olfactory > vigilance > foraging behaviors). The authors should try to weight the frequency of each behavior by the probability of detection that should be related with the relative speed of one beahviour to the others. Likely the picture that the authors can draw from weighted frequencies could be different to that reported now (see also lines 263-264).

iii) Even when the authors clearly stated that information from cameras placed at non-random points were not considered in the statistical models (but see below), they did not provide detailed information about the frequencies of each behavior at random vs non-random sampling points. I suggest including this information in order to complete the description of the spatio-temporal patterns in the behaviors. Likely some of the behaviors were only detected in non-random points or were more frequently detected in that points. At this respect, line 208 suggests that non-random cameras were just used to describe salt ingestion, right? If I am right, this is a relevant result meaning that random points are enough to describe the main behaviors and therefore monitoring aggregation points is not needed at least when the aim is more qualitative (behaviors’ description) than quantitative (estimating frequency of each behavior). Thus, comparative information between random vs non-random points is needed to enlarge discussion at this respect.

iv) Some discussion is lacked in relation to the results usefulness for implementing control strategies in the wild. This is announced in the abstract and in the introduction and, therefore, is expected to find further discussion at this respect. I recommend highlighting the applied component of their results in a new version of the manuscript.

Other considerations:

Lines 114-115. Authors described the minimum distance between cameras but information providing the effective distance is lacked. In relation to distance, it is probable that data from closer cameras can infringe the independence assumption of the observations. Can the authors provide information about the real distance among cameras? Was independence among observations tested or controlled in the statistical analysis? For this purpose, hierarchical models accounting for spatial autocorrelation can be an alternative to GAM (see for instance hSDM R package).

Lines 126-127. The effective area of detection for wild boar is quite small (see Hofmeester al. 2016 – Remote Sensing in Ecology and Conservation 3, 81-89) and it is not easily enlarged by locating cameras at higher heights. However, the camera height can drive differences in species detection depending of their size. Therefore, smaller individuals would be detected in low cameras in relation to tall (90cm height!) ones. Have authors observed some difference at this respect between age classes recorded at random and non-random points?

Line 136. More details about habitat characterization around cameras should be provided. For instance, what buffer size was considered to describe main species and cover? Since these are predictor variables in the models it is needed to provide a complete description of the procedure to obtain them.

Line 166. How were cameras operativity monitored? All cameras at the end of the experiment were operative (i.e. correctly positioned, with batteries, space in memory cards and taking pictures both at day and night)?

Line 168. Description of formula is not complete. It should be revised.

Line 199. Authors stated that locomotion was detected in 52% of video clips (number of video clips at random points 645). However, in supplementary material N=732 is reported for the locomotion model. Only the information at random points was actually used in the model?


Line 212. I am not sure if authors can refer to preferred habitats given their analytical framework. In my opinion to talk about avoided and preferred it is needed to account for habitat availability in the study area. Other case, results should be interpreted in terms of higher/lower frequency of video captures in relation to the habitat category used as reference in the statistical test.

Table 2. Decimal symbols should be dots. In addition, authors can include the difference obtained from Tukey's poshoc test in order to provide information about what is the behavior more frequent in the pair wise comparisons.

Reviewer 2 ·

Basic reporting

no comment

Experimental design

Your experimental design seem fine to me. I only have some minor points about the way of introducing the statistical approaches in the M&M section. In the attached pdf you will find some recommendations.

Validity of the findings

no comment

Additional comments

Dear Dana and Olivier,
I have found your work well written, timely and interesting.
My only concern is the lack of management implications in the discussion section. In fact, your started underlining the need of behavioural information for population management issues in the introduction section but later, in the discussion, you did not provide any kind of management implication or recommendation to alleviate the human-wildlife conflict caused by this wild pigs.
As you stated in the introduction section, wild boars are getting more and more common in urban areas (Castillo et al. 2019. Science of the Total Environment. 615: 282-288), cussing human conflicts. To my understanding, your behavioural recordings will help to minimize such conflicts, for example, minimizing the accumulation of garbage during periods when wild boars are actively looking for food. I believe you will find much more management implications that will be welcome by the readers and in turn will increase the impact of your research.

I hope that my comments will be useful for your investigation

Annotated reviews are not available for download in order to protect the identity of reviewers who chose to remain anonymous.

---

## Round 0.2 · Minor Revisions

Reviewer 2 accepted the publication but Reviewer 1 still questions the methods for comparing behaviors and requests additional analyses. If you disagree with some of the criticisms, you may write a rebuttal.

Reviewer 1 ·

Basic reporting

See comments for the authors

Experimental design

See comments for the authors

Validity of the findings

See comments for the authors

Additional comments

I acknowledge the efforts to accommodate some of my previous comments in the new version of the manuscript. In my opinion it now reads more clearly and can be more easily understood with the additional details included. However, some of the -major- methodological questions raised in the previous revision were not adequately handled:
1) The comparison among behaviours’ frequency without taking into account that faster behaviours have higher probability to contact with cameras and therefore to be overrepresented in the trapping rates is, at least, risky. Authors acknowledged this fact in the discussion as a potential explanation of their results, but they did not provide consistent reasons to do not consider it in the analyses. In my opinion, data obtained in this study allow to compare temporal trend within a given behaviour but comparisons in terms of frequency among behaviours should be cautiously considered.
2) The spatial autocorrelation in their data can bias the results in the statistical analyses (e.g. Sollmann 2018 – African Journal of Ecology 56, 740-749). Even when there are not overlap in the detection zones, the rates obtained by a given camera are expected to be closed to that obtained by its neighbors than to that for more far ones (i.e. spatial autocorrelation). I think that, at least, authors can test for spatial autocorrelation in the model residuals; if significant autocorrelation persists in the residuals then authors likely should move to another statistical framework accounting for spatial autocorrelation (e.g. hSDM or R-INLA R packages can be suitable options).

Reviewer 2 ·

Basic reporting

--

Experimental design

--

Validity of the findings

--

Additional comments

Congratulations

---

## Round 0.3 · accepted · Accept

I appreciate your effort and perseverance in revising the manuscript.